# Changes in the relationship between attachment and emotion recognition from adolescence to adulthood

Katalin Oláh ⬛ *

Institute of Psychology, Eötvös Loránd University, Budapest, Hungary

* olah.katalin@ppk.elte.hu

## Abstract

The present study investigated the relationship between attachment and emotion recognition in adolescents and adults. While previous research has found an association between these constructs in young children, as of yet, only a handful of studies have extended the investigation to later periods of life, either focusing on adults or adolescents. To capture the developmental changes that may take place during the transition from childhood to adulthood, we present findings with the same methodology on a group of adults and adolescents between the ages of 14 and 18. Participants completed the Reading the Mind in the Eyes test and filled out a self-report questionnaire on attachment style. Results suggest that attachment avoidance is associated with poorer emotion recognition capacities in adolescents, specifically in the case of positive emotions, while anxiety may have a positive, although weaker effect on mindreading. These associations are attenuated significantly by adulthood. Results are discussed in terms of information processing biases associated with attachment anxiety and avoidance.

## Introduction

Theory of mind, or mentalization, is a social cognitive capacity that has been extensively investigated in various fields of psychology, with a special emphasis on its origins and developmental trajectory. While significant advances have been made in the study of both typical and atypical populations (e.g., [1–4]), as of yet, little attention has been devoted to exploring interindividual differences within the healthy adult population. One factor that has been suggested to influence mentalization is the quality of attachment a person experiences in close relationships of which the origin can be traced back to infancy. Bowlby's [5] theory on attachment describes how early experiences with caregivers define the way individuals relate to social partners and social situations across the lifetime. A secure attachment formed in infancy means that the individual has learned that their social environment is safe and they can expect their needs to

**Data availability statement:** All datafiles are available from the OSF site of the project: https://osf.io/yw6xm/?view_only=bdf645d-934b544eb86a4c063fc2c7004

**Funding:** This research received funding from the National Research, Development and Innovation Office of Hungary under the grant agreement OTKA PD 143535 awarded to K.O. Funder website:https://nkfih.gov.hu/for-the-applicants The funders did not play any role in conducting the study or the preparation of the manuscript.

**Competing interests:** The author have declared that no competing interests exist.

be met by others. However, when the early attachment figure is absent, often unavailable or inadequately responds to the infant's needs, an insecure pattern of attachment develops and the infant will represent social partners as unreliable, unpredictable or even dangerous. Bowlby [6] claimed that these early experiences result in enduring schemata or scripts about socially relevant events and partners that he termed „*internal working models*". Ainsworth and colleagues [7] have extended Bowlby's theory by identifying three distinct attachment styles (secure, insecure-avoidant and insecure-ambivalent) in young children through their well-known procedure, the Strange Situation Test, to which later a fourth category, disorganized, was added [8].

Since internal working models are thought to contain schemata about the self and potential attachment figures as well as how attachment-related events are likely to unfold, it has been linked by various authors to a core social cognitive capacity: mentalization [e.g., [9,10]]. A number of studies have indeed confirmed that attachment security may be linked to performance on various theory of mind tasks in childhood (for a review see [11]). At the same time, theories have been put forth to explain this relationship. Since our study focuses on adolescents and adults, we will consider here two broader groups of accounts that are relevant to how this relationship may unfold later in life. On the one hand, from a developmental perspective, maternal behavior that promotes a secure attachment may also foster the development of mentalization skills – especially when it comes to understanding emotional states – as in this case, caregivers share and explore a wide range of emotions with their children and provide appropriate help with affect regulation [10,12,13]. At the same time, attachment security allows infants to engage in social interactions outside their immediate family as well which provides them with sufficient opportunity to learn about mental states and how they relate to overt behavior [10]. Thus, on this developmental view, theory of mind deficiencies can be mostly attributed to a lack of appropriate experience. If this is the case, then – in the absence of any serious conditions that would permanently bias information processing – a delayed development may be compensated for later in life through novel experiences with various social partners. A number of studies indeed suggest that the link between theory of mind and attachment may be attenuated as children grow [e.g., 13,14]

On the other hand, attachment characteristics may not simply limit the occasions to learn about mental states, but can introduce systematic distortions to the way socially relevant information is processed [15]. Bowlby [16] claimed that these biases can be best understood in the context of how an individual reacts to threats of potential psychological harm. Secure individuals' internal working models are created on the basis of mostly positive attachment-related experiences, making them more optimistic about the outcome of social interactions. At the same time, their representations of both the self and others are generally positive [17], which endows them with a sense of competence to cope with possible negative experiences. This makes secure individuals willing and able to deal with a wide-range of attachment-relevant information [15]. Insecure individuals, on the other hand, have generally negatively biased internal working models and develop so-called secondary attachment strategies [6] for affect-regulation, instead of proximity-seeking. One possible strategy,

deactivation (see, for example [18]), results in the exclusion of information related to attachment figures and social interactions, making insecurely attached individuals process less amount of information in general [15].

Mikulincer and Shaver [19] have proposed that depending on the specific type of insecure attachment, people develop predictably different defense or emotion regulation strategies that may influence mindreading. While adult attachment has been described by Hazan and Shaver [20] to follow a similar classification system as that of children ([7], another theoretical approach describes adult attachment in terms of two orthogonal dimensions that result in 4 types of attachment [17]. According to this view, everyone can be characterized by 1, the level of *anxiety* they experience in relationships and 2, their tendency to *avoid* closeness in relationships. Mikulincer and Shaver [19] proposed that high levels of anxiety leads to different information processing strategies than high levels of avoidance. People with high attachment anxiety assume proximity-seeking to be a viable protection from harm, yet fail to trust that it will be consistently available. As a result, anxiously attached individuals deploy a *hyperactivating* strategy whereby they constantly monitor information about the availability of potential attachment figures and become extremely vigilant in their relationships. Avoidant individuals' working models, on the other hand, lack schema that would suggest that proximity-seeking leads to the fulfillment of their needs and the alleviation of psychological pain, and thus learn to defend themselves by denying attachment needs altogether and to disregard attachment-relevant information, known in the literature as a *deactivating* strategy.

If attachment style indeed shifts social information processing in specific ways, then resulting difficulties (or benefits) in mindreading capacities may still be evident in later childhood and adulthood as well. As of yet, the role of attachment style in theory of mind processes in adolescence and adulthood has attracted considerably less interest in scientific research than its effect in childhood. Moreover, most studies focus on one specific type of mental state: emotions. Since attachment insecurity may fundamentally influence emotion regulation processes (see [19]) and mental contents involving emotions may be especially likely to be associated with potential psychological harm, it is reasonable to assume that this aspect of the mindreading system would be affected by attachment security. For example, Fraley and colleagues [21] have used a face-morphing paradigm to probe the hypothesis that attachment anxiety results in an increased vigilance to social information and thus leads to better performance in identifying emotional states. Their results suggest that attachment-related anxiety is associated with earlier detection of changes in emotional states that, in turn, may hinder correct identification of the emotion as individuals often prematurely arrive at a conclusion. However, when anxious individuals had the opportunity to form a judgment within the same timeframe as people with less anxiety, they proved to be more accurate.

Using the Reading the Mind in the Eyes test (henceforth, RMET [22]) Hünefeldt and colleagues [14] have presented somewhat contradictory findings. They investigated whether adolescents' emotion recognition was influenced by attachment anxiety or avoidance with different social partners. They found that anxiety with mother was negatively correlated with emotion recognition and this relationship was more pronounced for younger adolescents. Interestingly, in a group of young female adults, the opposite relationship was found between anxiety and the recognition of emotions that are neutral in valence or difficult to recognize [23], while a different research group failed to find any relationship between performance on the RMET and attachment styles [24].

Thus, to further clarify the role of attachment style in emotion recognition and how this relationship may change during the transition from childhood into adulthood, we administered the RMET together with a self-report questionnaire both to a group of adolescents (Study 1) and young adults (Study 2). Besides investigating two age groups with the same methodology, we also aimed to explore whether the valence of the given emotion to identify affects participants' success on the task. Different hypotheses may be formed about the role of emotional valence. If insecure individuals – especially with a predominant tendency for avoidance – defend themselves against psychological harm with a deactivating strategy [19], then identifying negative emotions, which they are trained to disregard, may be especially challenging for them. However, if – as Dykas and Cassidy [15] proposes – insecurity in attachment is not only linked to less accuracy in social information processing but to a negativity bias as well, then positive emotions may be more often missed both by anxious and avoidant individuals.

## Pre-test: Classification of the RMET stimuli based on valence

To classify the stimuli based on valence (negative, neutral and positive), we followed the methods of Harkness, Sabbagh, Jacobson, Chowdrey and Chen [25]. Considering that the language of the present study differs from those in which similar classifications have been done on the stimuli of the test, it was necessary to replicate the classification rather than use the one reported by Harkness and colleagues [25]. 14 adults ($M_{Age}$ = 36.42 years; $SD_{Age}$ = 7.77 years, 7 females) were presented with each picture and the corresponding word describing the emotion on separate slides and were asked to rate the emotional valence of the stimulus on a 7-point Likert scale (1 = very negative; 4 = neutral, 7 = very positive). The target word was positioned centrally below the picture and participants could navigate between the items at their own pace. We performed one-sample t-tests to determine if average scores on an item differed significantly from neutral (4). Those items that were significantly below or above 4, were classified as „negative" or „positive", respectively. Items where the mean of the ratings did not differ significantly from 4 were categorized as „neutral". The analysis yielded 11 positive, 16 negative and 9 neutral items. The list of positive, negative and neutral items can be found in the Supporting Information (S1 File). Data collection for the pre-test was conducted between 11 and 27 May, 2024.

## Study 1: Adolescents

In Study 1, we tested the potential relationship between adolescents' emotion recognition skills and attachment anxiety and avoidance. Participants filled out the Hungarian version of the Reading the Mind in the Eyes test ([22]; Hungarian adaptation: [26]) and the Attachment Scale Questionnaire (ASQ, [27]; Hungarian adaptation: [28]).

### Methods

**Ethics statement.** The research was approved by the Research Ethics Committee of Eötvös Loránd University, Faculty of Education and Psychology, Budapest, Hungary (no. 2023/339). All participants signed written informed consent at the beginning of the testing session. Parents were informed in writing about the study a week in advance of testing and were offered the possibility to decline the participation of their child. Written consent was only obtained from the participants as data was collected anonymously.

**Participants.** Participants were recruited from a secondary school in Budapest. An a priori sample size calculation using G*Power [29], assuming a moderate relationship (r = 0.3) between mentalization accuracy and attachment variables with an alpha level of 0.05 and two tailed-tests yielded a required sample size of 134. The final sample consisted of 143 adolescents between the ages of 14 and 18. 14 participants did not provide information about their gender; the rest of the sample was made up of 62 females and 67 males. Similarly, there were 12 missing data points regarding age, the mean age calculated for the available data was 15.29 years (SD = 1.53 years). An additional 13 adolescents were tested but excluded from analyses for missing or non-obvious responses on the RMET (n = 8) or more than 5 missing data points on the ASQ (n = 5). Recruitment announcement stated that individuals diagnosed with any psychiatric condition may not participate in this study.

**Materials.**

***Reading the Mind in the Eyes test*** [22]    In this test, participants are presented with black and white pictures of human faces cropped to only show the eye region of the face. Around the picture, four emotional states are listed, positioned in the four corners of the display. The participant's task is to choose the word that they think best describes the emotion conveyed by the photograph and to mark the correct response on a printed response sheet. The test was presented as a power point presentation that consists of one practice and 36 test trials, shown in a predefined order.

***Attachment Scale Questionnaire*** [27]    The Attachment Scale Questionnaire (ASQ) measures attachment through 40 items whereby participants rate attachment-related propositions on a 6-point Likert scale (1 = strongly disagree; 6 = strongly agree). The items of the ASQ have originally been classified into 5 subscales (Confidence; Discomfort with

Closeness; Need for Approval; Preoccupation with Relationships; Relationships as Secondary), grasping different aspects of attachment. In addition, the questionnaire may be used to measure the two broader dimensions, attachment anxiety and avoidance as well. During the Hungarian adaptation of the questionnaire [28], a 5 – factor structure emerged as well, however items were organized in a slightly different way into the following subscales: Importance of Relationships for the Self (IRS); Ambivalence, Distancing and Self-Deprecation (ADS); Confidence (CF); Self-Advocacy (SA); Dependency, Independency (DI). We chose the ASQ for the present study as it is appropriate to measure attachment not only in adults, but adolescents as well since its items do not have a strong focus on romantic relationships. To better explore how different aspects of attachment may be related to emotion recognition, we report analyses both with the 5 – factor model of the ASQ-H and the 2 – factor structure that captures anxiety and avoidance. Specifically, we used the factor structure identified by Karantzas and colleagues [30] to provide the best fit of the constructs. Both scales had good reliability in our adolescent ($\alpha_{Avoidance} = 0.847$; $\alpha_{Anxiety} = 0.888$) and adult sample ($\alpha_{Anxiety} = 0.863$; $\alpha_{Avoidance} = 0.843$) as well. The five subscales discovered during the Hungarian adaptation had acceptable reliability both in our adolescent (All $\alpha$: 0.677–0.854) and adult (All $\alpha$: 0.651–0.849) sample and were comparable to those found by Hámori and colleagues [28].

**Procedure.** Testing took place between 1 February and 15 June of 2024. Following the methods of Hünefeldt and colleagues [14], participants were tested in groups in their classrooms. The school principal and parents had been contacted online before testing sessions and had been provided with information about the study. During the testing session, participants were first briefly told about the goal of the research project and the general structure of testing. Then, they received a package with three documents in it: the informed consent, the response sheet of the RMET and a document containing the definitions of each word appearing in the test. Participants were asked to carefully read the informed consent and to sign it if they agreed to participation. Before starting the Reading the Mind in the Eyes test, they were asked to read through the response sheet and to look up any expressions that might be unfamiliar to them or of which they were unsure of the meaning. The RMET was presented on a projector in the form of a power point presentation that started with a slide with the instructions, followed by the practice trial and then the 36 test trials. Each picture was presented for 25 seconds, followed by a 5–second–long blank slide. After the RMET, participants were handed the ASQ which they could fill out at their own pace.

**Results and interim discussion.** For analyses, the mean scores of the ASQ scales and the following four variables for the RMET were used: 1, overall number of correct responses (RMETSum); 2, sum of correctly identified positive emotions (RMETPos); 3, sum of correctly identified negative emotions (RMETNeg); 4, sum of correctly identified neutral emotions (RMETNeut). We report all analyses both with the 2 – factor (Anxiety and Avoidance) model of ASQ as suggested by Karantzas and colleagues [29] and the 5 – factor model that emerged during the Hungarian adaptation [27]. We first conducted correlation analyses, then, to further clarify the relationship between the RMET scores, the attachment scales and the demographic variables in our sample, we fitted separate hierarchical linear regression models on the four RMET scores (RMETSum, RMETPos, RMETNeg and RMETNeut) with a forward stepwise method where sex and age were entered first and the attachment scales were added in the second step. An alpha level of 0.05 was used for all analyses.

Descriptive statistics of the scales can be found in Table 1. Shapiro-Walk tests showed that the following variables were not normally distributed: the positive (RMETPos; $W = 0.95$; $p < 0.001$), negative (RMETNeg; $W = 0.975$; $p = 0.01$) and neutral (RMETNeut; $W = 0.96$; $p < 0.001$) RMET items; Anxiety ($W = 0.977$; $p = 0.017$), ADS ($W = 0.968$; $p = 0.002$); Confidence ($W = 0.98$; $p = 0.034$); Self-advocacy ($W = 0.954$; $p < 0.001$) and Dependency, Independency ($W = 0.978$; $p = 0.021$). Therefore, the results of the non-parametric tests are reported for analyses including these variables. A preliminary test showed that girls outperformed boys on the aggregated score of the RMET (RMETSum) (M = 24.35 and 22.7, respectively; t(127) = −2.80; p = 0.006) and on the negative items (RMETNeg) as well (M = 10.95 and 9.81, respectively; U = 1352; p < 0.001). Moreover, girls (M = 3.79) manifested significantly higher levels of anxiety than boys (M = 3.18; U = 1321; p < 0.001). We found gender differences in the case of two scales of the 5 – factor model of ASQ: girls scored higher on the Importance

**Table 1. Descriptive statistics of the scales.**

| | Adolescents | | | | | Adults | | | | |
|---|---|---|---|---|---|---|---|---|---|---|
| | *M* | *SD* | α | *W* | *p (W)* | *M* | *SD* | α | *W* | *p (W)* |
| RMETSum | 23.5 | 3.38 | | 0.983 | 0.065 | 25.5 | 2.80 | | 0.979 | 0.029 |
| RMETPos | 7.39 | 1.74 | | 0.950 | <0.001 | 8.14 | 1.44 | | 0.945 | <0.001 |
| RMETNeg | 10.4 | 2.04 | | 0.975 | 0.01 | 11.4 | 1.89 | | 0.956 | <0.001 |
| RMETNeut | 5.77 | 1.56 | | 0.96 | <0.001 | 5.99 | 1.31 | | 0.947 | <0.001 |
| Anxiety | 3.47 | 1.03 | 0.888 | 0.977 | 0.017 | 3.23 | 0.9 | 0.863 | 0.986 | 0.164 |
| Avoidance | 3.22 | 0.745 | 0.847 | 0.992 | 0.645 | 3.10 | 0.732 | 0.843 | 0.990 | 0.377 |
| IRS | 3.82 | 0.932 | 0.854 | 0.987 | 0.195 | 3.67 | 0.785 | 0.810 | 0.991 | 0.522 |
| ADS | 2.87 | 0.960 | 0.841 | 0.968 | 0.002 | 2.73 | 0.936 | 0.849 | 0.966 | 0.001 |
| CF | 3.8 | 0.882 | 0.844 | 0.98 | 0.034 | 4.14 | 0.836 | 0.834 | 0.978 | 0.022 |
| SA | 2.47 | 0.85 | 0.703 | 0.954 | <0.001 | 2.27 | 0.736 | 0.651 | 0.971 | 0.004 |
| DD | 4.04 | 1.05 | 0.677 | 0.978 | 0.021 | 4.14 | 1.07 | 0.750 | 0.973 | 0.005 |

W represents test statistics for Shapiro-Wolf tests

of Relationships for the Self (IRS) scale ($M_{female}$ = 4.11; $M_{male}$ = 3.54; t(127)= −3.659; p<0.001), while the opposite pattern was observed on the Self-advocacy (SA) scale ($M_{female}$ = 2.29; $M_{male}$ = 2.6; U = 1449; p = 0.003). Moreover, age was significantly correlated with Dependency, Independency (ρ = 206; p = 0.018).

**Analyses with the 2 – factor model of the ASQ**   Correlation analyses showed that anxietjy was positively correlated with performance on RMETSum (ρ = 0.181; p = 0.03) and RMETPos (ρ = 0.229; p = 0.006). Thus, adolescents with higher levels of anxiety were better at identifying emotions in general and specifically, they were more skilled at identifying positive emotions.

The hierarchical linear regression analysis on RMETSum showed that the second model, containing the Avoidance and Anxiety scores accounted for 11% of the variance (F(4,122) = 3.79; p = 0.006), which was significantly higher than the variance explained by the first model ($\Delta R^2$ = 0.047; F(2,122) = 3.24; p = 0.043). Interestingly, including sex and age in the model revealed a different pattern of results with respect to Avoidance and Anxiety than the correlation analyses. There was only a marginally significant positive correlation between Anxiety and RMETSum (β = 0.166; t(122) = 1.813; p = 0.072) and a significant negative association emerged between Avoidance and RMETSum (β = 0.192; t(122) = −2.158; p = 0.033). The role of sex remained significant in the final model (β = 0.364; t(122) = 2.013; p = 0.046). In the case of positive emotions (RMETPos), the overall model accounted for 7% of the variance (F(4,122) = 2.179; p = 0.079) and Model 2 was a better fit than Model 1 ($\Delta R^2$ = 0.064; F(2,122) = 4.19; p = 0.017). Avoidance was negatively (β = −0.195; t(122) = −2.144; p = 0.034), while anxiety was positively (β = 0.225; t(122) = 2.392; p = 0.018) associated with the identification of positive emotions. The overall model on RMETNeg was significant ($R^2$ = 0.119; F(4,122) = 4.11; p = 0.004), however including the ASQ scales did not significantly increase the predictive value of the model (F(2,122) = 1.49; p = 0.229). Only gender of the participant emerged as a significant predictor (β = 0.549; t(122) = 3.055; p = 0.003), showing an advantage for girls. No significant predictors were found for neutral emotions (RMETNeut) and explained variance by Model 2 was low (3%).

**Analyses with the 5 – factor model of ASQ**   When the 5 – factor structure was applied to the ASQ, we found significant positive correlations between the Importance of Relationships for the Self (IRS) scale and RMETSum (r = 0.21; p = 0.012) and RMETPos (r = 0.238; p = 0.004). Self-advocacy was negatively correlated with RMETSum (ρ = −0.318; p<0.001), RMETPos (ρ = −0.240; p = 0.004), RMETNeg (ρ = −0.233; p = 0.005) and there was a marginally significant negative correlation with RMETNeut (ρ = −0.154; p = 0.065). Moreover, there was a marginally significant positive correlation between The Dependency, Independency scale and RMETPos (ρ = 0.145; p = 0.083) and a marginally significant negative correlation between the former and RMETNeg (ρ = −0.159; p = 0.058).

The hierarchical linear regression on RMETSum showed that the model including both age and sex and the ASQ scales accounted for 20% of the variance ($F(7,119) = 4.14$; $p < 0.001$) and Model 2 was a better fit than Model 1 ($\Delta R^2 = 0.132$; $F(5,119) = 3.91$; $p = 0.003$). Of the predictor variables, only Self-Advocacy was negatively associated with RMETSum ($\beta = -0.389$; $t(119) = -3.931$; $p < 0.001$). The overall model on RMETPos accounted for 16% of the variance ($F(7,119) = 3.331$; $p = 0.003$) and Model 2 was a better fit than Model 1 ($\Delta R^2 = 0.161$; $F(5,119) = 4.59$; $p < 0.001$). Self-Advocacy was negatively associated with RMETPos ($\beta = -0.355$; $t(119) = -3.519$; $p < 0.001$) and there was a marginally significant positive relationship with Importance of Relationships for the Self ($\beta = 0.205$; $t(119) = 1.811$; $p = 0.073$). The overall model on RMETNeg accounted for 14% of the variance ($F(7,119) = 2.68$; $p = 0.013$) but Model 2 was not a better fit than Model 1 ($\Delta R^2 = 0.039$; $F(5,119) = 1.08$; $p = 0.377$). This result was mirrored in the finding that only sex had a significant predictive value ($\beta = 0.527$; $t(119) = 2.879$; $p = 0.005$) with girls being more successful in identifying negative emotions than boys. The overall regression model on RMETNeut was only marginally significant and accounted for 10% of the variance ($F(7,119) = 1.98$; $p = 0.064$). Out of the predictor variables, Self-Advocacy was significantly negatively correlated with the target variable ($\beta = -0.259$; $t(119) = -2.483$; $p = 0.014$) while Importance of Relationships for the Self ($\beta = -0.223$; $t(119) = -1.907$; $p = 0.059$) and Dependency, Independency ($\beta = -0.190$; $t(119) = -1.765$; $p = 0.08$) had a marginal negative effect. Results are summarized in Table 2, additional information can be found in the Supporting Information (S2 and S3 Tables).

To summarize, the results suggest that avoidance and anxiety may differentially contribute to adolescents' emotion recognition skills: while those with higher levels of anxiety were marginally more successful in identifying emotions in general, avoidance was associated with worse overall performance on the test. This pattern of results was possibly driven specifically by the recognition of positive emotions where we found the same pattern. The results with the two-factor (Anxiety and Avoidance) model of ASQ were mostly corroborated by the findings with the 5-factor model; however, the discrepancies point out that avoidance may exert a more robust effect than anxiety. Except for negative emotions, we found that Self-Advocacy – a scale that contains items traditionally thought to describe avoidant tendencies – had a negative relationship with emotion recognition. The association between RMETSum and Importance of Relationships for the Self mirrors the similar finding with Anxiety as the IRS mostly contains items related to this construct; however, this result was only marginally significant. The analyses were consistent regarding the finding that the recognition of negative emotions was only affected by the gender of the participant, whereby girls outperformed boys.

This pattern of results fits best with the model proposed by Mikulincer and Shaver [19], suggesting that anxiety may be related to heightened vigilance when it comes to attributing mental states to others, whereas avoidance hinders mindreading capacities, possibly due to a defense mechanism that limits the processing of information that trigger the attachment system. Our results also suggest that the potential effect of attachment characteristics may be mostly manifested in the case of positive emotions.

## Study 2: Adults

### Materials and methods

**Ethics statement.** The research was approved by the Research Ethics Committee of Eötvös Loránd University, Faculty of Education and Psychology, Budapest, Hungary (no. 2023/338). All participants signed written informed consent at the beginning of the testing session.

**Participants.** 144 adults between the ages of 18 and 50 took part in the study (M = 22.9 years; SD = 5.61 years; 112 females, 31 males, 1 missing data). The majority of participants were below 25 years of age (for the distribution of participants by age and gender, see S6 Table). Participants were recruited either through university courses or with the help of student assistants with a snowball sampling method. As in the case adolescence, the lack of any psychiatric diagnosis was set as a precondition for applying to the study.

**Procedure.** The same questionnaires were administered to adult participants as adolescents. Participants were tested either in a group setting (n = 111) or individually (n = 33). In the former case, participants were tested in a university

**Table 2.** Results of the regression analyses on adolescents.

| | 2-factor Model of ASQ | | | | 5-factor Model of ASQ | | | |
|---|---|---|---|---|---|---|---|---|
| | β | R2 | ΔR2 | p(β/R2) | β | R2 | ΔR2 | p(β/R2) |
| **RMET Total (RMETSum)** | | | | | | | | |
| **Step 1** | | **0.063** | | 0.017 | | **0.06** | | 0.017 |
| Age | 0.055 | | | 0.526 | 0.064 | | | 0.456 |
| Sex | **0.364** | | | 0.046 | 0.266 | | | 0.134 |
| **Step 2** | | **0.111** | **0.047** | 0.006 | | **0.196** | **0.132** | <0.001 |
| Avoidance | **−0.192** | | | 0.033 | | | | |
| Anxiety | 0.166 | | | 0.072 | | | | |
| IRS | | | | | −0.015 | | | 0.894 |
| ADS | | | | | 0.163 | | | 0.224 |
| CF | | | | | −0.015 | | | 0.907 |
| SA | | | | | **−0.389** | | | <0.001 |
| DI | | | | | −0.093 | | | 0.363 |
| **RMET Positive Items (RMETPos)** | | | | | | | | |
| **Step 1** | | 0.003 | | 0.855 | | 0.003 | | 0.855 |
| Age | 0.066 | | | 0.456 | 0.036 | | | 0.678 |
| Sex | −0.134 | | | 0.471 | −0.24 | | | 0.185 |
| **Step 2** | | 0.067 | **0.064** | 0.075 | | **0.164** | **0.161** | 0.003 |
| Avoidance | **−0.195** | | | 0.034 | | | | |
| Anxiety | **0.225** | | | 0.018 | | | | |
| IRS | | | | | 0.205 | | | 0.073 |
| ADS | | | | | 0.107 | | | 0.435 |
| CF | | | | | 0.142 | | | 0.284 |
| SA | | | | | **−0.355** | | | <0.001 |
| DI | | | | | 0.148 | | | 0.157 |
| **RMET Negative Items (RMETNeg)** | | | | | | | | |
| **Step 1** | | **0.097** | | 0.002 | | **0.097** | | 0.002 |
| Age | −0.046 | | | 0.593 | −0.026 | | | 0.769 |
| Sex | **0.549** | | | 0.003 | **0.527** | | | 0.005 |
| **Step 2** | | **0.119** | 0.022 | 0.004 | | **0.136** | 0.039 | 0.013 |
| Avoidance | −0.143 | | | 0.107 | | | | |
| Anxiety | 0.087 | | | 0.344 | | | | |
| IRS | | | | | −0.036 | | | 0.758 |
| ADS | | | | | 0.024 | | | 0.861 |
| CF | | | | | −0.06 | | | 0.657 |
| SA | | | | | −0.159 | | | 0.124 |
| DI | | | | | −0.144 | | | 0.174 |
| **RMET Neutral item (RMETNeut)** | | | | | | | | |
| **Step 1** | | 0.03 | | 0.148 | | 0.03 | | 0.148 |
| Age | 0.11 | | | 0.228 | 0.137 | | | 0.129 |
| Sex | 0.247 | | | 0.192 | 0.338 | | | 0.179 |
| **Step 2** | | 0.03 | 0 | 0.43 | | 0.104 | 0.074 | 0.064 |
| Avoidance | −0.018 | | | 0.192 | | | | |
| Anxiety | 0.001 | | | 0.989 | | | | |
| IRS | | | | | −0.223 | | | 0.059 |
| ADS | | | | | 0.214 | | | 0.132 |

*(Continued)*

**Table 2.** (Continued)

| | 2-factor Model of ASQ | | | | 5-factor Model of ASQ | | | |
|---|---|---|---|---|---|---|---|---|
| | β | R2 | ΔR2 | p(β/R2) | β | R2 | ΔR2 | p(β/R2) |
| CF | | | | | −0.119 | | | 0.385 |
| SA | | | | | **−0.259** | | | 0.014 |
| DI | | | | | −0.19 | | | 0.08 |

IRS: Importance of Relationships for the Scale; ADS: Ambivalence, Distancing and Self-Deprecation; CF: Confidence; SA: Self-Advocacy; DI: Dependency, Independency

β represents the standardized coefficient.

Values in bold indicate significant effects at the level of 0.05.

β coefficients for sex and age are reported for the full model.

classroom, while in the latter, various – but always secluded and silent – locations were used (e.g., an empty dormitory room). The procedure of testing was identical to that described in Study 1. Testing participants in an individual setting followed the general procedure of the group setting with the exception that participants were presented with the RMET on a laptop screen. Testing session were completed between 1 December 2023 and 15 June 2024.

**Results and interim discussion.** Shapiro-Wilk tests revealed that none of the RMET scores were normally distributed (RMETSum: $W = 0.979$; $p = 0.029$; RMETPos: $W = 0.945$; $p < 0.001$; RMETNeg: $W = 0.956$; $p < 0.001$; RMETNeut: $W = 0.947$; $p < 0.001$). Similarly, the following ASQ scales failed to meet the criterion for normal distribution: Ambivalence, Distancing and Self-Deprecation ($W = 0.966$; $p = 0.001$); Confidence ($W = 0.978$; $p = 0.022$); Self-Advocacy ($W = 0.971$. $p = 0.004$) and Dependency, Independency ($W = 0.973$; $p = 0.005$). Therefore, the results of non-parametric tests are reported for analyses including those variables. Preliminary analyses revealed that males exhibited a marginally higher level of avoidance than females ($M_{male} = 3.30$; $M_{female} = 3.03$; $t(141) = 1.835$; $p = 0.069$). We found significant gender differences in the case of the Self-Advocacy ($M_{male} = 2.4$; $M_{female} = 2.2$; $U = 1147$; $p = 0.004$) and the Dependency, Independency scales ($M_{male} = 4.67$; $M_{female} = 4$; $U = 1225$; $p = 0.012$). Moreover, age was significantly negatively correlated with Anxiety ($\rho = −0.232$; $p = 0.005$) and Importance of Relationships for the Self ($r = −0.342$; $p < 0.001$), and there were marginally significant correlations between age and RMETNeg ($\rho = −0.153$; $p = 0.068$), and age and Ambivalence, Distancing and Self-Deprecation ($\rho = −0.141$; $p = 0.094$).

**Analyses with the 2-factor model of the ASQ** Correlation analyses revealed a negative relationship between Avoidance and RMETPos ($\rho = −0.191$; $p = 0.022$) and a marginally significant negative correlation between Anxiety and RMETPos ($\rho = −0.154$; $p = 0.065$).

Similarly to the analyses on adolescents, we fitted separate hierarchical linear regression models on the four RMET variables as targets, age and sex included in the first model and Anxiety and Avoidance added to the model in the second step. The model on RMETSum did not yield any significant predictors ($F(4,138) = 1.065$; $p = 0.376$). Moreover, the model containing the ASQ scales did not explain the target variable better than the one including only sex and age ($F(2,138) = 1.47$; $p = 0.234$). The model with RMETPos as dependent did not reveal any significant associations with the predictor variables either ($F(4,138) = 1.84$; $p = 0.125$). However, the model containing the ASQ scales was a significantly better fit than the one including only sex and age ($\Delta R^2 = 0.047$; $F(2,138) = 3.43$; $p = 0.035$). In the case of RMETNeg, Model 1 accounted for 5% of the variance ($F(2,140) = 3.98$; $p = 0.021$) and Model 2 was not a better fit than Model 1 ($F(2,138) = 0.37$; $p = 0.691$). The only significant predictor we found was age, showing a decline in the ability to correctly identify negative emotions with age ($\beta = −0.243$; $t(138) = 0.463$; $p = 0.006$). The model on RMETNeut did not yield any significant predictors ($F(4,138) = 0.715$; $p = 0.583$).

**Analyses with the 5-factor model of the ASQ** RMETSum and RMETPos were negatively correlated with Ambivalence, Distancing and Self-Deprecation ($\rho = −0.174$; $p = 0.037$ and $\rho = −0.267$; $p = 0.001$; respectively) and we found a

marginally significant negative correlation between RMETNeg and Importance of Relationships for the Self ($\rho = -0.147$; $p = 0.079$).

The hierarchical linear regression on RMETSum did not reach significance ($F_{(7,135)} = 1.038$; $p = 0.407$) and did not yield and significant predictors. The model including both the demographic variables and the ASQ scales accounted for 12,5% of the variance of RMETPos ($F_{(7,135)} = 2.753$; $p = 0.011$) and Model 2 was a better fit than Model 1 ($\Delta R^2 = 0.121$; $F_{(5,135)} = 3.75$; $p = 0.003$). The analysis revealed a moderate significant negative correlation between Ambivalence and RMETPos ($\beta = -0.433$; $t(135) = -3.416$; $p < 0.001$). RMETNeg was significantly, although weakly predicted by the model including sex and age ($R^2 = 0.05$; $F_{(2,140)} = 3.98$; $p = 0.021$), the only significant (negative) predictor was age ($\beta = -0.224$; $t(140) = -2.688$; $p = 0.008$). A marginally significant positive association was found between RMETNeut and Ambivalence ($\beta = 0.25$; $t(135) = 1.901$; $p = 0.059$), however, the model itself did not reach significance ($F_{(7,135)} = 1.16$; $p = 0.328$). Results of the regression analyses are summarized in Table 3, additional information on the models can be found in the Supporting Information (S4 and S5 Tables).

To sum up, findings with both the 5-factor and 2-factor models of ASQ suggest that attachment characteristics have little effect on emotion recognition in adulthood, but the 5-factor model was able to detect a negative association between the Ambivalence scale and the identification of positive emotions. This scale does not strictly represent typical markers of anxiety or avoidance but contains items that grasp an ambivalence towards relationships, originating from a low sense of self-worth, thereby relating to both attachment anxiety and avoidance with an emphasis on a tendency to withdraw from relationships.

## General discussion

In this study, we investigated the relationship between attachment characteristics and emotion recognition and how it may change during the transition from adolescence to adulthood. With this study, we set out to expand the current literature that has so far mostly focused either on young children or adolescents. Our methods were similar to the study conducted by Hünefeldt and colleagues [14] on adolescents as we used the Reading the Mind in the Eyes [22] test to measure emotion recognition skills and applied a self-report questionnaire on attachment style. However, since different questionnaires may grasp attachment in different ways, we decided to use the ASQ instead of the Experiences in Close Relationships [31] to additionally explore how well results with different measures would converge. The ASQ was selected specifically because it has been suggested to measure attachment appropriately not just in adults, but in adolescents as well [28]. Moreover, we tested both a 2 – and a 5 – factor model of the questionnaire to better understand how various classifications of attachment characteristics are related to emotion recognition.

We found that in the case of adolescents, scales corresponding to avoidance (the Avoidance scale of the 2 – factor model and the Self-Advocacy scale of the 5 – factor model) were negatively associated with performance on emotion recognition and there was a marginally significant positive correlation between scales related to anxiety and emotion recognition. The results show that when emotions are grouped by valence and analyzed separately, the effects only persist for positive emotions. A similar but weaker pattern was observed in adults as the regression models only showed a negative association between the identification of positive emotions and the Ambivalence scale of the 5 – factor model of the ASQ. The Ambivalence scale of the ASQ contains items that describe a tendency to withdraw from relationships and in that sense, this result may be taken to mirror the negative association between avoidance and RMETPos. However, this scale also strongly represents a decreased sense of self-worth which is traditionally considered to accompany attachment anxiety. During the Hungarian validation of the ASQ, Hámori and colleagues [28] highlighted that the Ambivalence scale grasps a complex facet of an individual's attitude towards relationships, therefore caution has to be made when relating this result to the one obtained with adolescents.

This pattern of results is most consistent with the idea proposed by Mikulincer and Shaver [19] that avoidance specifically hinders mindreading processes through a deactivating strategy, whereas anxiety may promote such processes. The

**Table 3. Results of the regression analyses on adults.**

| | 2-factor Model of ASQ | | | | 5-factor Model of ASQ | | | |
|---|---|---|---|---|---|---|---|---|
| | β | R2 | ΔR2 | p(β/R2) | β | R2 | ΔR2 | p(β/R2) |
| **RMET Total (RMETSum)** | | | | | | | | |
| **Step1** | | 0.009 | | 0.519 | | 0.009 | | 0.519 |
| Age | −0.107 | | | 0.23 | −0.102 | | | 0.26 |
| Sex | 0.15 | | | 0.477 | 0.223 | | | 0.304 |
| **Step2** | | 0.03 | 0.02 | 0.376 | | 0.05 | 0.042 | 0.407 |
| Avoidance | −0.022 | | | 0.815 | | | | |
| Anxiety | −0.139 | | | 0.147 | | | | |
| IRS | | | | | −0.106 | | | 0.313 |
| ADS | | | | | −0.199 | | | 0.134 |
| CF | | | | | −0.064 | | | 0.631 |
| SA | | | | | 0.078 | | | 0.418 |
| DI | | | | | 0.079 | | | 0.409 |
| **RMET Positive items (RMETPos)** | | | | | | | | |
| **Step1** | | 0.003 | | 0.783 | | 0.004 | | 0.783 |
| Age | 0.002 | | | 0.985 | 0.021 | | | 0.809 |
| Sex | 0.034 | | | 0.871 | 0.108 | | | 0.607 |
| **Step2** | | 0.051 | **0.047** | 0.125 | | **0.125** | **0.121** | 0.011 |
| Avoidance | −0.143 | | | 0.125 | | | | |
| Anxiety | −0.124 | | | 0.190 | | | | |
| IRS | | | | | −0.073 | | | 0.471 |
| ADS | | | | | **−0.433** | | | <0.001 |
| CF | | | | | −0.168 | | | 0.191 |
| SA | | | | | 0.007 | | | 0.943 |
| DI | | | | | 0.142 | | | 0.122 |
| **RMET Negative Items (RMETNeg)** | | | | | | | | |
| **Step1** | | **0.054** | | 0.021 | | **0.054** | | 0.021 |
| Age | **−0.243** | | | 0.006 | **−0.236** | | | 0.009 |
| Sex | 0.096 | | | 0.644 | 0.139 | | | 0.519 |
| **Step2** | | 0.059 | 0.005 | 0.077 | | 0.07 | 0.017 | 0.187 |
| Avoidance | 0.021 | | | 0.819 | | | | |
| Anxiety | −0.08 | | | 0.396 | | | | |
| IRS | | | | | −0.044 | | | 0.67 |
| ADS | | | | | −0.139 | | | 0.29 |
| CF | | | | | −0.063 | | | 0.634 |
| SA | | | | | 0.05 | | | 0.596 |
| DI | | | | | 0.077 | | | 0.414 |
| **RMET Neutral Items (RMETNeut)** | | | | | | | | |
| **Step1** | | 0.012 | | 0.345 | | 0.02 | | 0.345 |
| Age | 0.121 | | | 0.175 | 0.099 | | | 0.277 |
| Sex | 0.143 | | | 0.499 | 0.159 | | | 0.464 |
| **Step2** | | 0.02 | 0.005 | 0.583 | | 0.057 | 0.042 | 0.328 |
| Avoidance | 0.079 | | | 0.401 | | | | |
| Anxiety | −0.045 | | | 0.640 | | | | |
| IRS | | | | | −0.082 | | | 0.433 |

*(Continued)*

**Table 3.** (Continued)

| | 2-factor Model of ASQ | | | | 5-factor Model of ASQ | | | |
|---|---|---|---|---|---|---|---|---|
| | β | R2 | ΔR2 | p(β/R2) | β | R2 | ΔR2 | p(β/R2) |
| ADS | | | | | 0.25 | | | 0.059 |
| CF | | | | | 0.138 | | | 0.301 |
| SA | | | | | 0.086 | | | 0.371 |
| DI | | | | | −0.099 | | | 0.298 |

IRS: Importance of Relationships for the Scale; ADS: Ambivalence, Distancing and Self-Deprecation; CF: Confidence; SA: Self-Advocacy; DI: Dependency, Independency

β represents the standardized coefficient.

Values in bold indicate significant effects at the level of 0.05.

β coefficients for sex and age are reported for the full model.

latter result is also in line with the findings of Fraley and colleagues [21] who have shown that anxious individuals detect changes in emotions sooner and more precisely if given the same amount of time to visually observe a face. However, our data are contradictory to the results of Hünefeldt and colleagues [14] who have reported that emotion recognition was negatively correlated with anxiety in adolescents, specifically with anxiety with the mother and in early adolescence. This discrepancy may partly be due the differences between the self-report questionnaires used in the two studies. Hünefeldt and colleagues administered the Experiences in Close Relationships (ECR) which assesses anxiety and avoidance with different attachment figures separately whereas the ASQ contains more general propositions. Indeed, with the ECR, only anxiety with mother emerged as a significant (negative) predictor of emotion recognition, suggesting that the earliest attachment-related experiences may be especially relevant. Note that adult self-report questionnaires, such as the ASQ, grasp the individual's current feelings and attitudes towards relationships but do not define the origin of these. Most attachment theorists agree that while early experiences are crucial in shaping internal working models, subsequent attachment-related experiences can also shape an individual's attachment system [32]. Thus, it is possible that our study uncovered different effects than that of Hünefeldt and colleagues because the ASQ measures anxiety and avoidance in general terms, while the ECR can grasp differences between attachment figures. Importantly, while the abovementioned studies revealed significant effects regarding attachment anxiety, we were also able to show a negative relationship between avoidance and emotion recognition as well, a result that was more robust than the one about anxiety. While the fact that most effects found in our adolescent sample were no longer observable in adults suggests that social experiences accumulated throughout the lifetime attenuate the relationship between attachment and emotion recognition, the residual effects in adulthood may indicate that the early-originating tendency to avoid meaningful relationships continues to limit the opportunities to learn about mental states.

Importantly, our study highlights that these differences were most pronounced when identifying positive emotions. In the case of adolescents, anxiety was positively and avoidance was negatively associated with the identification of positive emotions and we also found a significant negative correlation with ambivalence toward relationships in adulthood. This pattern of results cannot be attributed to a general negativity bias as suggested by Dykas and Cassidy [15]. Alternatively, one may speculate about how early experiences with emotional displays could produce such a pattern of results. Mikulincer and Shaver [19] have suggested that anxiety in attachment may be caused by the tension created by the inconsistent availability of the caregiver, whereas avoidance is the result of consistently negative experiences of the caregiver's availability and behavior. Thus, since avoidant individuals' childhoods may especially be lacking in experiences with displays of positive emotions, the identification of those may be selectively impaired. In theory, hypervigilance caused by the inconsistent behavior of the caregivers of anxious individuals could result in better recognition of both negative and positive emotions. In fact, if the function of hypervigilance is to detect potential harm, then one might argue that the effects

would be mostly manifested in the case of negative emotions. Although this is speculative, it is possible that any potential advantage caused by hypervigilance may be masked by feelings of anxiety triggered by pictures that are not unambiguously positive, which interfere with performance. Such an interference may be caused, for example, by a tendency to avoid deeper processing of the negative or neutral stimuli and to respond to such pictures faster (see [21]). Future studies may test these hypotheses. Nonetheless, this result is in line with a finding that attachment anxiety is selectively associated with better mindreading performance after a positive attachment-related mood induction [33]. Importantly, we would like to point out that hypervigilance as an emotion regulation strategy does not necessarily have to result in across the board better mindreading skills. As a mechanism, it can foster mindreading through an increased amount of attention directed towards social cues, however, emotion recognition skills are also largely influenced by the input one receives throughout the lifetime (i.e., the emotional displays presented by social partners, how the individual's emotions have been mirrored by others [34], etc.).

While we did not find considerable differences when applying the 2 – and 5 – factor models of ASQ, the results corroborate Hámori and colleagues' [28] suggestion that the 5 factors of the Hungarian questionnaire cannot simply be matched to the two broader categories of attachment characteristics (anxiety and avoidance). This was highlighted by the finding that in adults, the Ambivalence scale emerged as a significant predictor which incorporates items related to both anxiety and avoidance. Moreover, while we found significant predictors with both structures, the regression models with five scales seemed to fit the data better, as these models generally accounted for a higher percentage of the variance of the target variable.

It should be noted that the generalizability of the results with the adult sample may be handled with caution due to the uneven gender distribution. Namely, as the result of the recruitment process, the adult sample consisted of mostly females, while the distribution was more balanced in the adolescent sample and gender differences have been found in the case of multiple variables tested. While the regression analyses include sex and age and thus control for these effects, the current sample size did not allow us to fully investigate how gender and age may come into interaction with each other or the target variables.

Our study aimed to replicate previous findings on adolescents as well as to paint a more nuanced picture of the relationship between attachment and one particular facet of theory of mind, emotion recognition, beyond early childhood. Our findings are mostly consistent with the view that anxiety and avoidance result in different routes of social information processing [19] and that early disadvantages may be successfully compensated for as a result of experience. However, note that correlational studies such as ours cannot define the causal relationship between theory of mind and attachment. These two constructs may shape each other in complex ways throughout development (e.g., [10;35]). In addition, it is also reasonable to assume a role of latent variables that shape both constructs at the same time (see [14]). Finally, note that emotion recognition grasps only a very specific aspect of theory of mind, an otherwise complex social cognitive capacity. In the case of adolescents and adults, only a few sporadic studies have explored this relationship, focusing on various types of mental states [21,24,36–38]. Future studies should systematically investigate how different types of mental states and mindreading processes are modulated by attachment characteristics.

## Supporting information

**S1 File. List of neutral, negative and positive items of the RMET.**
(PDF)

**S2 Table. Results of the regression analyses on adolescents with the 2 – factor model of the ASQ.**
(DOCX)

**S3 Table. Results of the regression analyses on adolescents with the 5 – factor model of the ASQ.**
(DOCX)

**S4 Table.  Results of the regression analyses on adults with the 2 – factor model of the ASQ.**
(DOCX)

**S5 Table.  Results of the regression analyses on adults with the 5 – factor model of the ASQ.**
(DOCX)

**S6 Table.  Distribution of adult participants by gender and age group.**
(DOCX)

## Acknowledgments

I thank Rita Bencsik, Anna Ferenczi, Blanka Szóráth, Borbála Láncz, Hanna Lakits and Flóra Léna Dobó for their help with data collection.

## Author contributions

**Conceptualization:** Katalin Oláh.

**Data curation:** Katalin Oláh.

**Formal analysis:** Katalin Oláh.

**Funding acquisition:** Katalin Oláh.

**Investigation:** Katalin Oláh.

**Methodology:** Katalin Oláh.

**Project administration:** Katalin Oláh.

**Resources:** Katalin Oláh.

**Supervision:** Katalin Oláh.

**Visualization:** Katalin Oláh.

**Writing – original draft:** Katalin Oláh.

**Writing – review & editing:** Katalin Oláh.

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
