## [Decision Letter · Decision Letter 0]

12 Mar 2025

PONE-D-24-36706Changes in the relationship between attachment and emotion recognition from adolescence to adulthoodPLOS ONE

Dear Dr. Oláh,

Thank you for submitting your manuscript to PLOS ONE. After careful consideration, we feel that it has merit but does not fully meet PLOS ONE’s publication criteria as it currently stands. Therefore, we invite you to submit a revised version of the manuscript that addresses the points raised during the review process.

We look forward to receiving your revised manuscript.

Kind regards,

Sanjoy Kumer Dey, M.D

Academic Editor

PLOS ONE

Journal Requirements:

2. Please report in the Methods section the day, month and year of the start and end of the recruitment period for each study. Please note that if this information is not included when your manuscript is resubmitted, it may be rejected.

Please provide additional details regarding participant consent for study 2. In the ethics statement in the Methods, please ensure that you have specified (1) whether consent was informed and (2) what type you obtained (for instance, written or verbal, and if verbal, how it was documented and witnessed). If the need for consent was waived by the ethics committee, please include this information

Reviewers' comments:

Reviewer's Responses to Questions

**Comments to the Author**

1. Is the manuscript technically sound, and do the data support the conclusions?

Reviewer #1: Yes

Reviewer #2: Yes

2. Has the statistical analysis been performed appropriately and rigorously? 

Reviewer #1: Yes

Reviewer #2: Yes

3. Have the authors made all data underlying the findings in their manuscript fully available?

Reviewer #1: Yes

Reviewer #2: Yes

4. Is the manuscript presented in an intelligible fashion and written in standard English?

Reviewer #1: Yes

Reviewer #2: Yes

5. Review Comments to the Author

Reviewer #1: This is a behavioral study investigating the relationship between attachment and emotion recognition in adolescents and adults.

This is an interesting study and I only have a few minor comments to help clarify a few aspects of the study.

In the abstract, “we present findings …” should be rephrased as the study is not just on “young adults and adolescents” as the age range is up to 50 years old.

In my opinion, the introduction is too long and expands too much about early childhood and infants which makes it confusing since it is not the purpose of the study. But I leave that to the author and editor to decide whether to shorten it.

Did the author consider the interaction between age and sex as a predictor (and if not, why not?)? Especially in the adult group. Also, there is a large discrepancy in sample size between males and females in the adult group. Could that reflect a bias in recruitment? This should be acknowledged as a limitation, or discussed as a potential explanation of differences between the adolescents and adults. In this context, was age correlated with the different variables tested? A supplementary Figure displaying the distribution of sex by age might be informative.

Is there any information about mental health / neurodevelopmental disorders for the participants? Were any participants autistic (for instance) as it could significantly impact their capacity to identify emotions. This information should be clarified.

In S1_File, in the neutral emotion section, cautious is listed twice.

Reviewer #2: The manuscript addresses an important avenue of research and the analyses are robust. The results will contribute to the literature on relationships between attachment and emotion recognition in different developmental stages. I commend the authors.

6. PLOS authors have the option to publish the peer review history of their article (what does this mean? ). If published, this will include your full peer review and any attached files.

**Do you want your identity to be public for this peer review?** For information about this choice, including consent withdrawal, please see our Privacy Policy .

Reviewer #1: No

Reviewer #2: **Yes: ** Ali Evren Tufan

---

## [Author Response · Author response to Decision Letter 1]

25 Apr 2025

Responses to Reviewer 1

In the abstract, “we present findings …” should be rephrased as the study is not just on “young adults and adolescents” as the age range is up to 50 years old.

Thank you for the suggestion, the adjustment has been made.

In my opinion, the introduction is too long and expands too much about early childhood and infants which makes it confusing since it is not the purpose of the study. But I leave that to the author and editor to decide whether to shorten it.

I agree with the reviewer that the first part of the introduction was lengthy, therefore I have shortened it somewhat (see paragraph 1). However, in order to shed some light on how attachment may play a role in adult theory of mind and emotion recognition specifically, it is important to understand the mechanisms that may contribute to this potential link. Since attachment characteristics are rooted in early experiences, I believe it is necessary to discuss these potential mechanisms from a developmental point of view. Therefore, the sections that present how biases in information processing may appear as a result of attachment insecurity have not been deleted or shortened.

Did the author consider the interaction between age and sex as a predictor (and if not, why not?)? Especially in the adult group.

I thank the reviewer for the suggestion. Indeed, interaction terms would be interesting to add to the analyses; however, it is unfortunately not possible with the current dataset as detecting interactions in the regression models requires a significantly larger sample size (for discussions on this see for example Leon, A. C., & Heo, M. (2009). Sample Sizes Required to Detect Interactions between Two Binary Fixed-Effects in a Mixed-Effects Linear Regression Model. Computational statistics & data analysis, 53(3), 603–608. https://doi.org/10.1016/j.csda.2008.06.010 or https://statmodeling.stat.columbia.edu/2018/03/15/need16/

). In addition, I have run a power analysis for the interaction term with standard parameters (effects size: 0.2; alpha=0.05.; power=0.8, 1 covariate (age) and 2 groups (for sex)), and it resulted in a required sample size of 416 which is signficantly higher than my own. Thus, this particular study is underpowered to reliably test these effect. Nonetheless, I trust that in spite of this, the reviewer will find that the current results are still sound enough to publish - with acknowledging the limitations, of course. To ensure clarity on this issue, I have extended the discussion (see next point also).

I would also like to point out that although the age range is rather wide, the majority of the participants fall at the lower end of the age scale (see also below).

Also, there is a large discrepancy in sample size between males and females in the adult group. Could that reflect a bias in recruitment? This should be acknowledged as a limitation, or discussed as a potential explanation of differences between the adolescents and adults.

Indeed, recruitment differed for the adult and the adolescent samples. Adolescents were recruited in high schools and were tested in their own classrooms, thus gender distribution here largely resembles that of the classes themselves. Adults, on the other hand, were recruited through a university course for credit. These participants were thus university students. Although recruitment did not limit the faculties from which students may have applied (with the only exception that psychology students could only participate if they were in their freshman year), with this method of recruitment, we generally end up with an uneven distribution of females and males. The discussion has been extended to reflect on this issue (lines 555-561.).

In this context, was age correlated with the different variables tested? A supplementary Figure displaying the distribution of sex by age might be informative.

Thank you for the suggestion, another Supplemantary file (S6) have been added to show the distribution of participants by age and gender. Moreover, both for adolescents and adults zero-order correlations involving age have been calculated and reported in the preliminary analysis sections (lines 270-271 and 400-403).

Is there any information about mental health / neurodevelopmental disorders for the participants? Were any participants autistic (for instance) as it could significantly impact their capacity to identify emotions. This information should be clarified.

During recruitment, we explicitly stated that a lack of any psychiatric diagnosis was a precondition for participation. Since it would be difficult to unambiguously define conditions that do not affect such social cognitive skills, we felt it was necessary to extend this constraint to any diagnosis. A sentence has been added to the methods section to make this explicit in the manuscript (lines 192-193).

In S1_File, in the neutral emotion section, cautious is listed twice.

Some of the expressions may appear multiple times in the file since the classification for valence was done on each item by presenting both the verbal expression and the picture. Since there are some items in the RMET with the same emotion expression as the correct choice, these appear separately in this document.

Response to Reviewer 2

I am grateful for the kind words on our work and the manuscript.

---

## [Decision Letter · Decision Letter 1]

9 May 2025

Changes in the relationship between attachment and emotion recognition from adolescence to adulthood

PONE-D-24-36706R1

Dear Dr.Katalin Oláh 

We’re pleased to inform you that your manuscript has been judged scientifically suitable for publication and will be formally accepted for publication once it meets all outstanding technical requirements.

Kind regards,

Sanjoy Kumer Dey, M.D

Academic Editor

PLOS ONE

Additional Editor Comments (optional):

Reviewers' comments:

Reviewer's Responses to Questions

**Comments to the Author**

1. If the authors have adequately addressed your comments raised in a previous round of review and you feel that this manuscript is now acceptable for publication, you may indicate that here to bypass the “Comments to the Author” section, enter your conflict of interest statement in the “Confidential to Editor” section, and submit your "Accept" recommendation.

Reviewer #1: All comments have been addressed

2. Is the manuscript technically sound, and do the data support the conclusions?

Reviewer #1: Yes

3. Has the statistical analysis been performed appropriately and rigorously? 

Reviewer #1: Yes

4. Have the authors made all data underlying the findings in their manuscript fully available?

Reviewer #1: Yes

5. Is the manuscript presented in an intelligible fashion and written in standard English?

Reviewer #1: Yes

6. Review Comments to the Author

Reviewer #1: (No Response)

7. PLOS authors have the option to publish the peer review history of their article (what does this mean? ). If published, this will include your full peer review and any attached files.

**Do you want your identity to be public for this peer review?** For information about this choice, including consent withdrawal, please see our Privacy Policy .

Reviewer #1: No

---

## [Editor Report · Acceptance letter]

PONE-D-24-36706R1

PLOS ONE

Dear Dr. Oláh,

I'm pleased to inform you that your manuscript has been deemed suitable for publication in PLOS ONE. Congratulations! Your manuscript is now being handed over to our production team.

Kind regards,

on behalf of

Dr. Sanjoy Kumer Dey

Academic Editor

PLOS ONE